# An Experimental Study of the Crystallinity of Different Density Polyethylenes on the Breakdown Characteristics and the Conductance Mechanism Transformation under High Electric Field

**DOI:** 10.3390/ma12172657

**Published:** 2019-08-21

**Authors:** Liwei Zhou, Xuan Wang, Yongqi Zhang, Peng Zhang, Zhi Li

**Affiliations:** 1School of Electrical and Electronic Engineering, Harbin University of Science and Technology, Harbin 150080, China; 2Key Laboratory of Engineering Dielectrics, Application Harbin University of Science and Technology, Harbin 150080, China

**Keywords:** polyethylene, crystallinity, breakdown strength, conduction mechanism, SCLC, field strength

## Abstract

In order to study the crystallinity of different density polyethylenes, this paper conducts an experimental study on the transformation of the conductance mechanism under a high electric field. In this experiment, X-ray diffraction (XRD), differentials scanning calorimetry (DSC), direct current (DC) breakdown of low-density polyethylene (LDPE), linear low-density polyethylene (LLDPE), medium-density polyethylene (MDPE) and high-density polyethylene (HDPE), as well as the conductivity characteristics under an electric field of 5–200 kV/mm are tested. In addition, the electric field–current density curves of the four kinds of polyethylene are fitted to analyze their conductance transition in non-ohmic regions under different high field strengths, through applying the mathematical formula of a variety of conductance mechanisms. The experimental results are as follows: as the density of polyethylene increases, the crystallinity increases continuously. Moreover, the continuous increase of crystallinity causes the electric conduction flow under the same field strength to decrease significantly. The field strength corresponding to the two turning points in the conductance characteristic curve increases simultaneously, and the breakdown field strength increases accordingly; through analysis, it is found that in the high field, as the electric field increases, the conductance mechanism develops from the ohmic conductance of the low field strength region to the bulk effect of the high field strength region (Poole–Frenkel effect). Then, it develops into the electrode effect to the high field strength (Schottky effect), although the threshold field strength of this conductance mechanism transition increases with the increase of crystallinity.

## 1. Introduction

Polyethylene is a partially crystalline solid, whose properties are highly dependent on the relative content of the crystalline phase and amorphous phase, i.e., crystallinity. Polyethylene is polymerized from monomeric ethylene. It is widely applied in the insulating material of power cables due to its symmetrical molecular structure without polar groups, which gives it excellent electrical and mechanical properties [1,2]. According to different polymerization methods, polyethylene can be classified into linear low-density polyethylene (LLDPE), low-density polyethylene (LDPE), medium-density polyethylene (MDPE), and high-density polyethylene (HDPE). LLDPE possesses a regular short-chain structure, and, although its crystallinity and density are similar to those of LDPE, the intermolecular force is larger. The macromolecules of LDPE have many branches, which cannot be closely and regularly arranged with each other. Moreover, its branching degree is high. Medium-high density polyethylene is a linear macromolecule with low branching degree and regular structure [3].

As an insulating material, polyethylene can easily cause electrical trees under the effect of a high electrostatic voltage field, which will eventually lead to insulation breakdown [4]. Currently, global scholars generally agree that the space charge effect plays an important role in the insulation aging process [5,6]. The existing research has mainly utilized space charge limited current (SCLC), Schottky effect, Poole–Frenkel effect and hopping conductivity to explain the conductance mechanism of pure polyethylene or polyethylene nanocomposites in high field strength regions (non-ohmic regions). Some scholars believe that the conductivity mechanism of polyethylene under high field strength is not dominated by a single conductance mechanism [7,8,9], but a variety of conductance mechanisms [10]. Others believe that the charge transport mechanism of doped nano-polyethylene in a high field strength region (non-ohmic region) is dominated by ion hopping conductance, which can be deduced from the formula that the ion jump distance increases with increasing temperature [11,12]. They also note the “pre-electric stress” effect of polymer nanocomposites under a high electric field [13,14]; other research indicates that it is dominated by electronic hopping conductance, and the jump distance decreases with increasing temperature [15].

Many scholars have undertaken a significant amount of research on the conductance property of polymer nanocomposites. However, most have only studied the modification of low-density polyethylene matrix nanocomposites [16,17,18]. There are very few studies on the conductance property of polyethylene with different densities. In addition, due to the complexity of the structure of polymer materials, there is currently a lack of sufficient understanding of the conductance mechanism of polyethylene. Based on the existing research on polyethylene and polyethylene nano-polyethylene [19,20], this paper studies the effects of different crystallinity on the DC breakdown strength as well as the conductance property of different density polyethylene. It also discusses the effects of crystallinity of polyethylene with different densities on its conductance property based on XRD and DSC.

## 2. Experiment

### 2.1. Experimental Materials

The paper selected linear low-density polyethylene (LLDPE), low-density polyethylene (LDPE), medium-density polyethylene (MDPE) and high-density polyethylene (HDPE) as test materials. The LLDPE model was LLDPE7042 with a density of 0.922 g/cm^3^. The LLDPE was produced by Jilin Petrochemical Company (Jilin, China). The model of LDPE was LD200BW with a density of 0.918 g/cm^3^, and the model of MDPE was MDPE157 with a density of 0.935 g/cm^3^. LDPE and MDPE were produced by Sinopec Beijing Yanshan Branch. The HDPE model was DMDA 8008 with a density of 0.954 g/cm^3^. HDPE was produced by Daqing Petrochemical Company (Daqing, China).

### 2.2. Sample Preparation

The LDPE sample was pressed by a plate vulcanization machine (Harper Electric Company of Harbin, Harbin, China) at 110 °C. The process was as follows: preheated for 5 min before pressurization, then pressurized 5 MPa every 5 min, and finally pressed at a maximum pressure of 15 MPa. The thickness of the sample utilized to test the DC breakdown was 60 μm, and the thickness of the sample utilized to test the conductance was 50 μm. The samples of LLDPE, MDPE, and HDPE were prepared in the same manner. However, the pressing temperature was changed to 150 °C. All the samples were made by depositing aluminum as electrodes on the vacuum membrane plate machine. Finally, the prepared electrode samples were placed in a vacuum drying oven for 24 h with 50 °C short-circuit temperature.

### 2.3. DSC Test

The thermal properties of polyethylene with four different densities were tested by differential scanning calorimetry (METTLER TOLEDO in Zurich, Switzerland). During the experiment, 0.005–0.007 g samples were weighed and placed in aluminum crucibles, protected by high-purity nitrogen. Moreover, the heating and cooling temperature rate were set to 10 °C/min. The sample was first heated to 200 °C to completely melt, eliminating the influence of thermal history. Subsequently, it was cooled to 40 °C to obtain a crystallization process curve, and then heated to 200 °C to obtain a melting curve.

### 2.4. XRD Test

Phase analysis of polyethylene with four different densities was carried out using an X-ray diffractometer (X’pert) produced by Philips, Amsterdam, The Netherlands. The X-ray source was CuK α; the tube voltage was set to 40 kV; the tube flow was set to 40 mA; the phase analysis was performed in the *θ*–2*θ* scanning mode; the step size was set to 0.05°; the time constant was 1 s. The *θ*–2*θ* scanning mode was applied for fine scanning, the step size was set to 0.02°, and the time constant was 20 s.

### 2.5. Conductivity Test

A schematic diagram of the electrical conductivity test system is shown in Figure 1. A Keithley 6517B electrometer/high resistance meter (Tektronix of Johnston, Ohio, USA) (minimum theoretical measurement accuracy 10–15 A) and a self-made three-electrode system (measuring electrode diameter 25 mm) were utilized in a vacuum environment (the pressure inside the vacuum box was 0.1 MPa). Pure LLPE, LDPE, MDPE, and HDPE with a thickness of 50 μm were measured. The conductivity of the field strength of 5–200 kV/mm was 30 min. Based on the LabView system, a computer accomplished the automation as follows: automatic control of the boost voltage; automatic data acquisition and storage of the high-voltage DC power supply, the 6517B high-resistance meter, and the controller; automatic control of the protection circuit. The voltage source had a step of 250V each time. For 30 min, when the pressurization time reached 30 min, the controller controlled the high-voltage relay to operate, and the 6517B high-impedance meter was connected to the measurement loop for a 30-second current test, collected 10 data per second, and finally selected the average value as the voltage. After measurement, the sample was short-circuited for 25 min. Although the literature indicates that the current steady-state needs to be maintained for several hours or several days [21], it was found during the test that the current basically reached steady state during the process of pressurizing or depressurizing for 25 min. Thus, the influence of the previous low field strength and the pressurized residual charge on measurement under the subsequent high field strength is ignored. Subsequently, the controller controlled the action of the high voltage vacuum. The relay operated to continuously disconnect the 6517B high-resistance meter, and then pressurized again to repeat.

## 3. Experiment Results and Discussion

### 3.1. Polyethylene Melting Characteristics

Figure 2 reveals the melting characteristics of four kinds of polyethylene. It can be seen from Figure 2 that the melting temperature of LDPE is the lowest and the melting peak is small; the melting temperature of LLDPE is higher than that of LDPE, and the melting peak is lower than that of LDPE; the melting temperature of MDPE is higher than that of LLDPE, and the melting peak is higher than that of LLDPE; andthe highest melting temperature and melting peak of HDPE are very similar to those of MDPE, and the melting peak area is slightly larger than that of MDPE. Crystallinity (*X_c_*) can be utilized to characterize the ratio of the crystalline part from the semi-crystalline polymer. The calculation of *X_c_* is as shown in Equation (1).
(1)$$X_{c}=\frac{\Delta H_{m}}{\Delta H_{100}}\times 100\%$$
where, Δ*H_m_* is the enthalpy absorbed by the test sample during the heating process, and Δ*H*_100_ is the enthalpy absorbed by the sample during the crystallization-melting process. The Δ*H*_100_ of polyethylene is 293 J/g [22,23]. The crystallinity for four kinds of polyethylene was calculated by DSC test software. The results are shown in Table 1. It can be seen from Table 1 that the crystallinity from four kinds of polyethylene is different: the crystallinity of LDPE is the lowest; the crystallinity of HDPE is the highest; the crystallinity of MDPE is higher than that of LLDPE; and the crystallinity of LLDPE is slightly higher than of LDPE, although both are similar. It can also be seen that LLDPE and LDPE not only have similar density, but also have similar crystallinity. Meanwhile, it is known from experiments that the crystallinity increases with the density.

### 3.2. Analysis of the Crystal Structure of Different Polyethylene

Figure 3 shows that the four kinds of polyethylene with different densities have obvious diffraction peaks at nearly the same position 2*θ* = 21.42°, indicating that these four kinds of polyethylene have typical crystal structures. Moreover, from the image and data, it can be concluded that the diffraction peak intensities for four kinds of polyethylene are arranged from large to small: HDPE, MDPE, LLDPE, and LDPE. The crystallinity of the polymer is directly proportional to the diffraction peak intensity of the XRD. Hence, it can be concluded that the crystallinity of the four polymers is in the order (from large to small): HDPE, MDPE, LLDPE, and LDPE. The image results of XRD are completely consistent with the results obtained through the DSC data calculation. In this way, the crystallinity for four kinds of polyethylene with different densities can be determined in this experiment.

### 3.3. DC Breakdown Strength Test

Figure 4 is a Weibull distribution of DC breakdown strength for the four kinds of polyethylene with different densities. The shape parameters and characteristic breakdown strengths for the four kinds of polyethylene are shown in Table 2. The shape parameter indicates the dispersion of the breakdown data, and the characteristic breakdown strength represents the electric field strength when the overall sample reaches a 63.2% breakdown probability. It can be seen from Table 2 that the breakdown strength for the four kinds of polyethylene is different, and that the breakdown strength of LDPE is the lowest. The breakdown field strength of HDPE, MDPE, and LLDPE is 37.96%, 28.56% and 15.01% higher than that of LDPE, respectively. It can be concluded that the breakdown strength for the four kinds of polyethylene increases with the increase of crystallinity, mainly because the free volume of polyethylene becomes smaller as the crystallinity increases. Thereby, the free path of electrons is reduced, it becomes difficult for them to accumulate energy in the electric field, and the probability of electrons accelerating under the electric field is lowered so that the breakdown strength is correspondingly increased [24,25,26,27]. It can also be seen from Table 2 and Figure 4 that the larger the shape parameter, the smaller the dispersion of the breakdown strength data. The polyethylene exhibits more stable dielectric properties.

### 3.4. Conductivity Test

#### 3.4.1. Conduction Current Theory

(a) Traps and space charge

There are many localized states in the forbidden band gap of the polymer. These localized states can capture the carriers in the material to form a space charge, which acts as a trap. Therefore, the localized state is also called a trap. Space charge is sometimes referred to as trapped charge. Traps are mainly caused by crystal imperfection, caused by structural defects or impurities, or both. Traps are roughly divided into two types, mainly formed by structural defects or chemical defects. It is generally believed that discrete trapping levels are associated with chemical impurities doped in the lattice, and quasi-continuous trapping levels are related to imperfections of crystal structure [25].

(b) Space charge limited current (SCLC)

The current-voltage characteristics of the dielectric comply with Ohm’s law at low electric fields, i.e., at the beginning of pressurization. When the voltage (or electric field intensity) reaches a certain value *U_Ω_* (or *E_Ω_*), the concentration of the injected carriers will increase, accumulating a large amount of space charge and causing the space charge limited current. In this way, the current flowing through the dielectric is transformed from the region of ohmic current to the region of space charge limited current. When the field strength applied to the material exceeds the breakover field strength, a large accumulation of carriers and space charge limited current will occur in the insulation, which may cause various aging conditions. Thus, the breakover voltage *U_Ω_* (or electric field intensity *E_Ω_*) is sometimes referred to as the electric degradation threshold of the dielectric material, which is the ideal situation without traps. There are inevitably various traps for the actual dielectric materials as mentioned above. When there are traps, the trapping of the injected charge causes the breakover voltage to be much larger than when there is no trap, and the current is made smaller. As the voltage *U* applied to the material increases, the amount of injected charge increases, and the traps in the material are gradually filled. When the voltage reaches a certain value *U_m_*, the traps are filled, and the injected electrons will no longer cause trapping so that the current in the insulation increases so sharply that it turns to the region of space charge limited current without traps. At this time, the density *J* of the space charge limited current follows Calder’s law with trap filling or without traps, as shown in Figure 5 [26].

The expression of the density of the space charge limited current is as follows:(2)$$J=\left(\frac{9}{8}\varepsilon_{r}\varepsilon_{0}\mu \frac{U^{2}}{d^{3}}\right)\theta$$where *ε*_0_ is the vacuum dielectric constant, *ε_r_* is the relative dielectric constant, *μ* is the permeability, *d* is the dielectric thickness, *θ* is the control parameter of the trap. *Θ* = *n*/(*n* + *n_t_*), which is the ratio of the free carrier concentration to the total carrier concentration. *n_t_* is the trapped carrier concentration, and *n* is the free carrier concentration; since *n_t_* >> *n*, *θ* ≈ *n*/*n_t_*, usually *θ*
*≤* 10^−7^ [24].

The breakover voltage *U_Ω_* of the space charge limited current can be expressed as Equation (3), and *n_t_* can be expressed as Equation (4):(3)$$U_{\Omega }=\frac{8end^{2}}{{9\varepsilon }_{r}\varepsilon_{0}\theta }$$where *e* is the amount of electron charge.

(4)$$n_{t}=\frac{9\varepsilon_{r}\varepsilon_{0}U_{\Omega }}{8ed^{2}}$$

Equation (5) can be obtained after taking the logarithm of both sides of Equation (2):(5)$$\ln J=\ln\frac{9\varepsilon_{r}\varepsilon_{0}\mu }{8d^{3}}+2\ln U$$

It can be seen from Equation (5) that the current density and the applied voltage of the SCLC is linear in the double logarithmic coordinate with a slope of 2.0.

#### 3.4.2. Test Results of Conduction Current

Figure 6 and Figure 7 show the *E*-*I* characteristic curves of the electrically conductive flow for four different density polyethylenes of LDPE, LLDPE, MDPE and HDPE, as well as the logarithmic curves in the lg*I*-lg*E* coordinate system. The data is segmented in Figure 7. It can be seen from the fitting results of Figure 7 that the conductivity curves for the four kinds of polyethylene have two turning points (namely, point A and point B), and three areas (namely, T1, T2, and T3), and four different density clusters. The field strengths corresponding to the A and B points of polyethylenes are given in Table 4. The slopes of the fitted straight line of the T1, T2, and T3 regions are given in Table 3. In addition, according to the *E*-*I* curve of the four different density polyethylenes in Figure 6, we know that the electric conductivity values of LDPE, LLDPE, MDPE, and HDPE, respectively, decrease in the same field strength, so the conductivity is in the same field strength. The next is also reduced in turn. The carrier mobility is related to the migration barrier and the jump distance. As the density of polyethylene increases, the thickness and convergence of polyethylene platelets increase [27], and the molecular chains of MDPE, HDPE, LLDPE, and LDPE are tight. The degree of compaction is weakened, and the compactness of the platelets is weakened in turn. Therefore, the height of the migration barrier is sequentially decreased. The jump distance is related to the thickness of the platelet. As the crystallinity increases, the thickness of the platelets increases, and the average jump distance increases. Hence, the rate of the carrier migration is significantly reduced as the crystallinity of the polyethylene increases.

According to the data in Table 3, the slope of the T1 region, i.e., the low field strength region (ohmic region), is approximately 1 and remains substantially unchanged, while the slope of the T2 region, i.e., the high field strength region (non-ohmic region), increases with density. The height gradually decreases, but the slope values are all greater than 2; the slope of the LDPE in the T3 region is close to 2, and the slopes of the remaining three clocks are close to 1.

According to Table 4, the corresponding field strength at the first turning point A in the conductance characteristic curve of LDPE, LLDPE, MDPE, and HDPE in the lg*I*-lg*E* coordinate system is gradually increased. Moreover, it can be seen that with the crystallization of polyethylene, the degree gradually increases, and the electric field threshold from the ohmic zone to the non-ohmic zone gradually increases. Many scholars hold that there are many local states in the forbidden band gap of polymers. These local states can trap carriers in materials to form space charges [7]. According to space charge limiting current theory, from the ohmic region to the space charge limiting current region, the transition voltage of the current region corresponds to the electric field strength at which the space charge begins to accumulate. The cause of the above phenomenon may be that the LDPE electrical conduction flow is the largest and the most effective carrier, and that of LLDPE, MDPE and HDPE is decreased in the four materials at the same field strength in the ohmic region. Therefore, LDPE may first start the accumulation of space charge. The field strength corresponding to the turning point in the high field strength region (non-ohmic region), namely, the point B, is gradually increased, and the field strength corresponding to B point is continuously increased as the crystallinity of the polyethylene increases. Many scholars believe that the turning point in the high field strength region (non-ohmic region) may be due to the tunneling effect under the charge filling area or high field strength [28].

## 4. Discussion

### 4.1. High Field Slope Change

We generally believe that the non-ohmic current of a solid dielectric in a high electric field mainly appears as a space charge limited current (SCLC), which is mainly due to the rapid proliferation of carrier concentration with the enhancement of the electric field. Both the electrode effect and the bulk effect cause nonlinear propagation of carriers in the medium, causing the carrier concentration in the medium to become a function of the electric field strength. Therefore, the high electric field non-ohmic current characteristics may have two phenomena of electrode characteristic control and body characteristic control [21,24,29]. Since the solid dielectric is low in bulk conductivity, it is generally believed that the transport process of carriers in the ohmic conductance region of the weak electric field is dominated by the hopping conductance of the impurity ions [7,12,24], and the current belongs to the bulk limiting process. However, when the conditions (temperature, electric field) change, the high-field non-ohmic conductance will change from the body effect limit to the electrode effect limit, and the high electric field download stream transport process belongs to the amorphous state [30,31]. In theory, the slope of the space charge limited current zone should be 2 in the double logarithmic coordinates. From the measurement results of the non-ohmic region of the T2 region of Figure 7, it can be seen that the slope of the high field strength curve of the four different density polyethylene materials ranges from 2 to 4 regardless of the voltage form, and the slope gradually decreases with the increase of crystallinity. The slope of the high field strength conductance curve for polyethylene and its nanocomposites is reported to vary from 2 to 5 [7,31,32]. Montanari studied the conductance mechanism of different kinds of polyethylene. It is believed that the slope of the curve in the logarithmic coordinates of the high field strength region is greater than 2, and there is a space charge limiting current effect [29]. Moreover, the slope of the high field strength region cannot be applied to judge the quality of the material, which can only be utilized for the difference in the conductivity mechanism of the material [22]. In the T3 region, the electric field strength is further increased to a maximum of 200 kV/mm. According to Figure 5, we can see that the slope value of the T3 region is supposed to be 2 according to the theory. However, according to Table 3, we can see that the results obtained in the experiment are as follows: only the slope of LDPE in the T3 region is about 2.132, which is very close to 2, while the slopes of LLDPE, MDPE and HDPE in the T3 segment are close to 1. This phenomenon may be due to the deeper traps in the dielectric material when the field strength is particularly high and excited. Thus, when the field strength reaches the electric field corresponding to the second inflection point in the conductance characteristic curve, the carrier may be trapped by the deeper trap, which leads to the phenomenon that the slope of the conductance characteristic curve of the T3 region is much smaller than that of the T2 region.

### 4.2. High Field Conductance Theory

Currently, the proliferation process of carriers in the high electric field region is dominated by Schottky effect (electrode effect) and Poole–Frenkel effect (body effect) [33], and the charge transport mechanism is dominated by electron jump conductance [34]. In order to study the change of the conductivity mechanism of the four different density polyethylenes under high field strength at room temperature (25 °C), the mechanism of the above two high field strengths is briefly introduced.

(1) For the Schottky effect, the current density is controlled by the carriers injected into the medium by thermal excitation. The formula is as follows:(6)$$j=AT^{2}\cdot \exp[-\frac{\varphi -\sqrt{e^{3}E/4\pi \varepsilon_{0}\varepsilon_{r}}}{kT}]$$
where $\varepsilon_{r}$ is the relative dielectric constant; $\varepsilon_{0}$ is the vacuum dielectric constant; *A* is the Richardson–Dushman constant; φ is the work function of the metal.

It can be seen from Equation (5) that at high field strength, if the conductance mechanism is subject to the Schottky effect, the conductance characteristic curve will be linearly distributed in the ln*j*-*E*^1/2^ coordinate system [28].

(2) For the Poole–Frenkel effect, the conductance change can be described as:(7)$$\frac{j}{E}=\sigma =\sigma_{0}\cdot \exp\left[\frac{\sqrt{\frac{e^{3}E}{\pi \varepsilon_{0}\varepsilon_{r}}}}{kT}\right]$$
where *σ* is the volume conductivity; $\sigma_{0}$ is the initial conductivity; is the Boltzmann constant.

It can be seen from Equation (6) that at high field strength, if the conductance mechanism is subject to the Poole–Frenkel effect, the conductance characteristic curve is linearly distributed in the ln*j*-*E*^1/2^ coordinate system [29].

### 4.3. Data Segmentation Fitting Result

The carrier propagation of polyethylene in the high field strength region is a process from the limitation of body effect to the limitation of electrode effect. There are many interactions of conductive mechanisms [10,21], which cannot be fitted by a single conductive mechanism [7,21,32]. It can be seen from Figure 7 that the current characteristics of the four different density polyethylenes in the high field strength region are not dominated by the space charge limiting current. If polyethylene is synergistically acted by a variety of conductance mechanisms in the high field strength region (non-ohmic region), for the two carrier propagation processes (Schottky effect or Poole–Frenkel effect) in the high field strength in Section 4.2, the distribution in the ln*j*-*E*^1/2^ coordinate system may have no obvious law. However, if a certain conductivity mechanism plays a dominant role in a fixed temperature range and a certain field strength, the corresponding data in the range will exhibit a linear distribution in the ln*j*-*E*^1/2^ coordinate system. In order to verify the above reasoning, the data of the four different density polyethylenes in the high field strength region (non-ohmic region) are plotted in the ln*j*-*E*^1/2^ coordinate system, and the Poole–Frenkel effect and Schottky effect are more intuitively reflected. After leading conduction mechanism, conducting segmentation fitting of the image, and setting polyethylene relative permittivity *ε_r_* = 2.35, we select the best intercept and compare it in the ln*j*-*E*^1/2^ coordinate system when the slope is fixed. The result is shown in Figure 8.

From Figure 8, we can see as follows: the fitted line of the Poole–Frenkel effect is in the low field strength range of the high field strength region; the degree of coincidence with the experimental data is higher; and the fitting line with the Schottky effect is lower. When the field strength rises above a certain threshold, the slope of the experimental data decreases, which reduces the fit of the fitted line with the Poole–Frenkel effect, and the coincidence with the fitted line of the Schottky effect is significantly improved. The fitting results show that during the field strength increase, the conductance mechanism of the high field strength non-ohmic region changes from the Poole–Frenkel effect to the electrode (Schottky) effect. In addition, in Table 5, we can see that as the crystallinity of different density polyethylene increases, the threshold field strength of the conductance mechanism transition also increases, because the high electric field download stream transport process belongs to the local state in the amorphous region. The carrier transport mechanism under high electric field is dominated by electron hopping conductance [30,31]. The higher the crystallinity of polyethylene, the smaller the free volume and the smaller the free travel of electrons. Therefore, the field strength corresponding to the occurrence of the electrode limiting effect is higher.

## 5. Conclusions

In this paper, the experiments based on XRD, DSC, DC breakdown, and conductance characteristics of 5–200 kV/mm for four different density polyethylenes (LDPE, LLDPE, MDPE, and HDPE) at room temperature (25 °C) are measured. The conclusions are as follows:

(1) The crystallinity for the four types of polyethylene in different densities increases with increasing density.

(2) With the increase of crystallinity, the corresponding breakdown field strength of four kinds of polyethylene with different densities also increases greatly. Simultaneously, we find that the larger the shape parameter, the smaller the dispersion of the breakdown strength data. In this respect, polyethylene exhibits more stable dielectric properties.

(3) In the test of conductivity characteristics from four different densities of polyethylene, the electric field threshold of the ohmic region (T1 region) corresponding to LDPE, LLDPE, MDPE, and HDPE to the non-ohmic region is gradually reduced. Meanwhile, the field strength corresponding to turning point B of the non-ohmic zone is also gradually decreased. Its mechanism may be that the conductivity of the four materials gradually decreases, and the number of carriers decreases correspondingly under the same field strength. Therefore, the carriers in large numbers may be the first to reach the turning point.

(4) In the image of the lg*I*-lg*E* coordinate system, after segmentation fitting, the slopes of the conductivity curves of the four materials in the T1 region are very close to 1, which is almost the same as the theoretical value; the slope value range of the T2 region is 2-4, indicating that there is an SCLC effect in this region; and the slope of the conductance characteristic curve of the T3 region is much lower than that of the curve in the T2 region, which may be because the region is not filled with traps. Since the further improvement of the electric field strength applied in the non-ohmic region stimulates the deeper trap of the dielectric material, the carrier again enters the trapping process, resulting in the slope of the T3 region being smaller than the slope of the conductance characteristic curve of the T2 region.

(5) The conductivity mechanism for the four different densities of polyethylene in the high field has a transition process from the Poole–Frenkel effect to the Schottky effect. Moreover, with the increase of the polyethylene’s crystallinity, the threshold of field strength corresponding to the transformation of the conductivity mechanism increases gradually.

## Figures and Tables

**Figure 1 materials-12-02657-f001:**
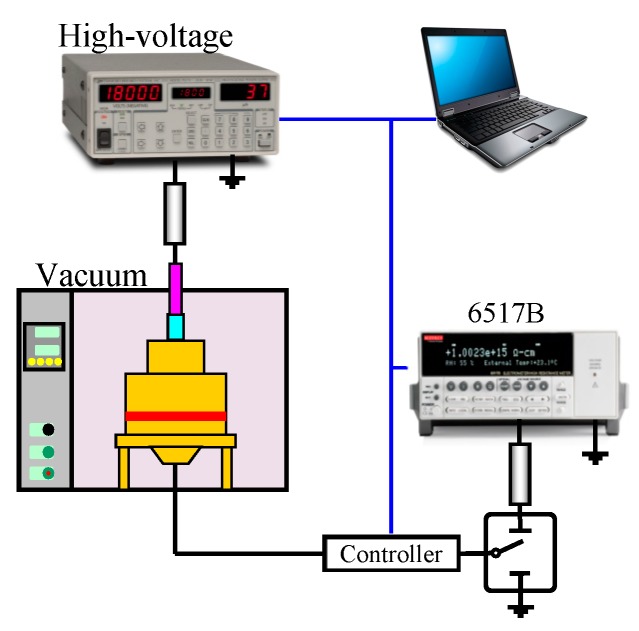
Test schematic.

**Figure 2 materials-12-02657-f002:**
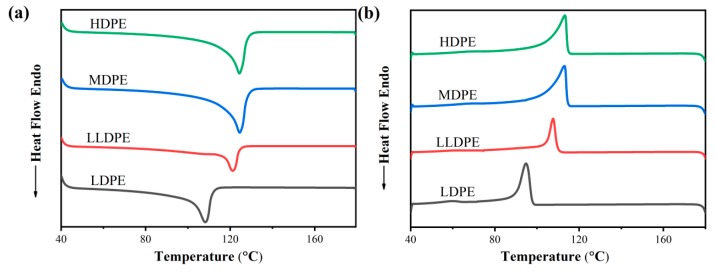
(**a**) Differentialscanning calorimetry (DSC) cooling curvesand (**b**) heating curves of four kinds of polyethylene.

**Figure 3 materials-12-02657-f003:**
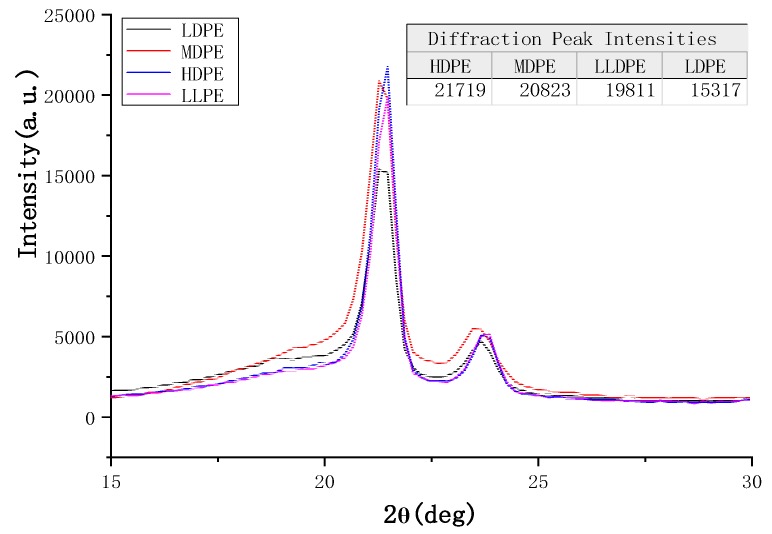
X-ray diffraction (XRD) images of four kinds of polyethylene.

**Figure 4 materials-12-02657-f004:**
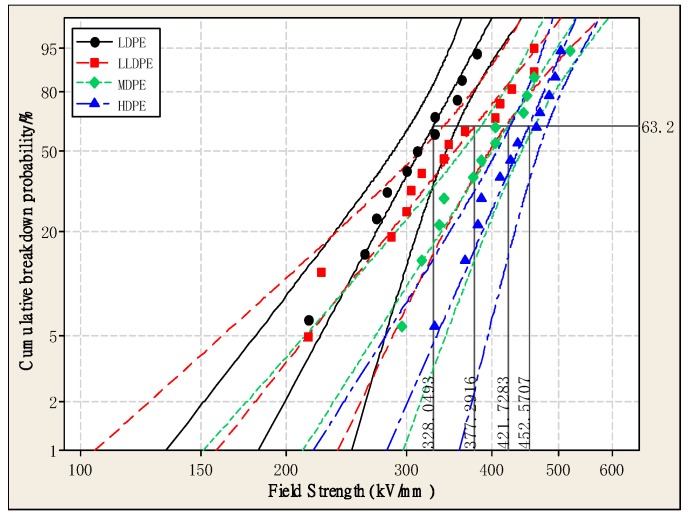
Weibull plots of the DC breakdown strength of four kinds of polyethylene.

**Figure 5 materials-12-02657-f005:**
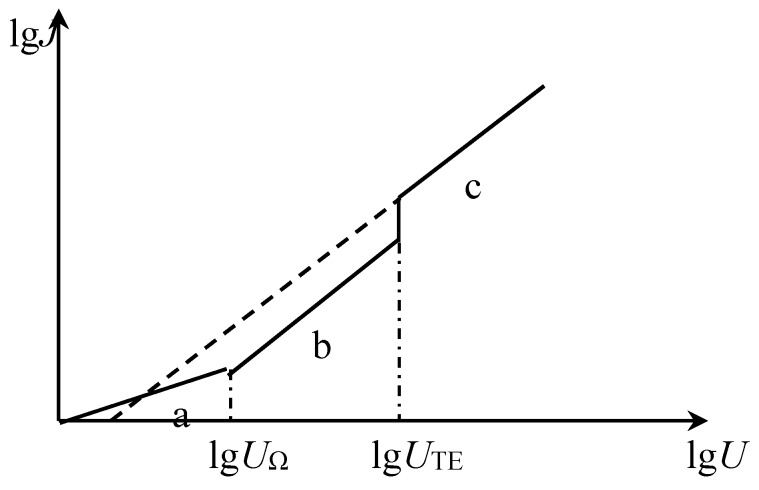
Relationship between space charge limited current in dielectrics and the applied voltage. a-region: linear region, i.e., ohmic conduction current region; b-region: Calder’s law region of space charge limited current when trapped; c-region: Calder’s Law region with trap filling or without traps.

**Figure 6 materials-12-02657-f006:**
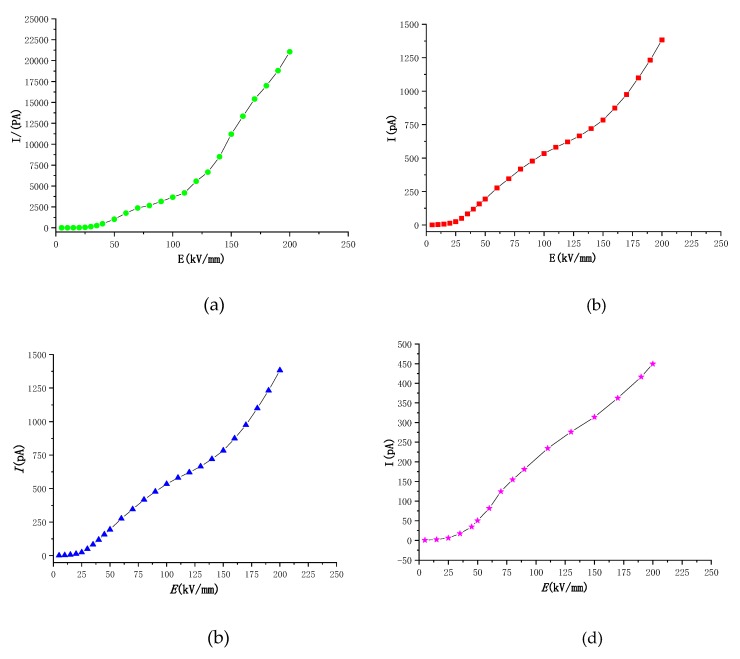
Conduction current characteristic curve of polyethylene with different densities: (**a**) LDPE I-E; (**b**) LLDPE I-E; (**c**) MDPE I-E; (**d**) HDPE I-E.

**Figure 7 materials-12-02657-f007:**
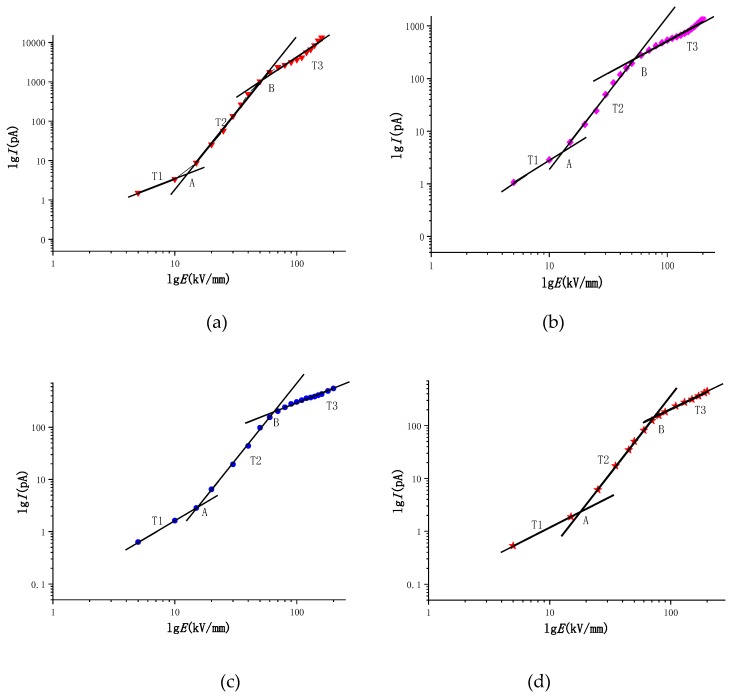
Logarithmic form fitting diagram of electrical conductivity flow of four different density polyethylenes: (**a**) LDPE lgI-lgE; (**b**) LLDPE lgI-lgE; (**c**) MDPE lgI-lgE; (**d**) HDPE lgI-lgE.

**Figure 8 materials-12-02657-f008:**
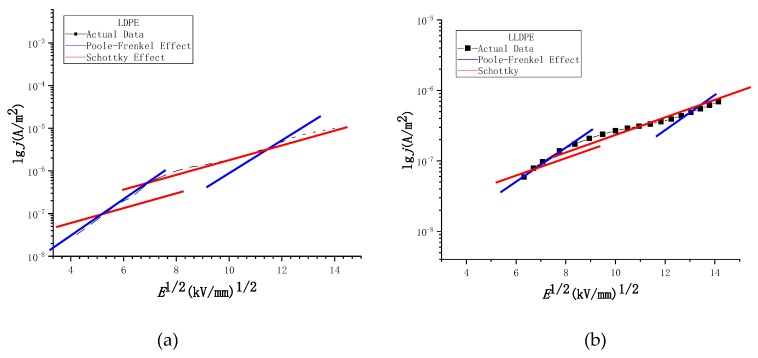
Fitting curve of the four kinds of polyethylene in ln*j*-*E*^1/2^ coordinates: (**a**) Fitting curve of the LDPE in lnj-E1/2 coordinates; (**b**) Fitting curve of the LLDPE in lnj-E1/2 coordinates; (**c**) Fitting curve of the LDPE in lnj-E1/2 coordinates; (**d**) Fitting curve of the LDPE in lnj-E1/2 coordinates.

**Table 1 materials-12-02657-t001:** Melting process parameters of four kinds of polyethylene.

Sample	T_m_ (°C)	T_c_ (°C)	Δ*H_m_* (J/g)	(*X_c_*) %
LDPE	108	94	113.5	38.73
LLDPE	121	107	115.6	39.45
MDPE	124	113	141.7	48.36
HDPE	125	113	149.9	51.17

**Table 2 materials-12-02657-t002:** Parameter and breakdown strength of polyethylene.

Material	Shape Parameter	Characteristic Breakdown Strength (kV/mm)
LDPE	7.791	328.0
LLDPE	5.277	377.3
MDPE	6.654	421.7
HDPE	9.577	452.6

**Table 3 materials-12-02657-t003:** The slope of the line in different areas of different materials.

Different Areas	T1	T2	T3
LDPE	1.16	3.94	2.13
LLDPE	1.42	2.93	1.23
MDPE	1.36	2.89	0.89
HDPE	1.15	2.83	1.12

**Table 4 materials-12-02657-t004:** The corresponding field strength at different turning points of different materials.

The Turning Point	LDPE	LLDPE	MDPE	HDPE
A	12.76 kV/mm	13.33 kV/mm	15.68 kV/mm	18.37 kV/mm
B	50.88 kV/mm	53.61 kV/mm	65.34 kV/mm	71.89 kV/mm

**Table 5 materials-12-02657-t005:** Field strength of the four different density polyethylenes in the high field strength region from body effect to electrode effect.

Temperature/(°C)	Threshold Field Strength of LDPE/(kV/mm)	Threshold Field Strength of LLDPE/(kV/mm)	Threshold Field Strength of MDPE/(kV/mm)	Threshold Field Strength of HDPE/(kV/mm)
25	44.89	54.76	72.25	77.44

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
