# Peer review of "An Experimental Study of the Crystallinity of Different Density Polyethylenes on the Breakdown Characteristics and the Conductance Mechanism Transformation under High Electric Field"

_materials, 2019, doi:10.3390/ma12172657_

Round 1

Reviewer 1 Report

attached. 

Author Response

First of all, I would like to thank all the professional reviewers for your comments. All the authors are very grateful for this. We have made the following changes according to the criticism and suggestions of the professional reviewers. The modified sections are covered with a yellow background in the article.

Reviewer 1:

Extensive English editting is required. Current version is not acceptable.

Answer: It has been modified, all statements in the entire paper have been reworked and retouched

Details of materials used should be provided including vendors,melt index (or molecular weight), and features in chemical structures: For example, what is the kind of branches in LLPE.

Answer: It has been modified, the chemical model of the material and the manufacturer of the material have been relabeled.

Detail interpretation for XRD peaks are missing. Figure 3 is not clearly visible for each sample.An enlarged figures in a range of 10 to 40 2theta degree are necessary.

Answer: It has been modified , figure 3 The XRD curves. of the four polyethylenes are drawn separately.

There are only four sets of data with too many figures. Some can be deleted to make the manuscript succinct. Figure 6 should be deleted as log-log plots are adequate. Figure 5 and 7 can be combined into one figure. Labels for sub-figures should be inserted like (a), (b), (c)…

Answer: It has been modified ,Figure 6, Figure 7, and Figure 8 are respectively re-typed. The presence of Figure 6 can more clearly show the conductance of different dielectrics in high field.

Check the sample code in Table 5.

Answer: It has been modified

Reviewer 2 Report

In this paper author study the crystallization of different density polyethylenes. Authors studied also the conductance mechanism under high electric field employing direct current (DC). Different techniques such as X-ray diffraction (XRD), different scanning calorimeter (DSC) were used to characterized the crystallization of low-density (LDPE), linear low density (LLDPE), medium density (MDPE) and high-density (HDPE) polyethylene. This manuscript show interesting data and should be published after minor revision.

Minor revision:

Please use the same nomenclature to name polyethylene, sometime you use “low-density “ sometimes “middle density” please revise whole manuscript. Please specify supplier of the materials in 2.1. Experimental materials Section Please specified the equipment which you use to perform DSC measurement. Please specify the equipment which you use to perform XDR measurement. Why you perform your DSC from 40 ºC In order to avoid reader confusion please remove the stabilization period from DSC curves. Please used the common standard programs to work with DSC and XDR. Use the same letter for DSC legend inside the graph. Please draw all 4 axis as in more published papers; applied for every Figure. Additionally, in DSC Figure mark the endo process with arrow. Please see related papers. Heating flow is arbitrary units and the values should be remove from the graph. Please avoid double information. You have melting point and crystalization point at Table and in Figure 2. Please graph Figure 3 from 10 to 70º in this way we will better appreciate the changes for each polyethylene. In present way is difficult to distinguish between each curve. In the case of Figure 6 please make the 4 graphs smaller (two in the same column and to in the same line (see different published papers). In this case readers with be easer compare your results obtained for each polypropylene. Make the same changes for Figure 7 and Figure 8.

Author Response

Reviewer 2 : Minor revision:

Please use the same nomenclature to name polyethylene, sometime you use “low-density “ sometimes “middle density” please revise whole manuscript. Please specify supplier of the materials in 2.1. Experimental materials Section Please specified the equipment which you use to perform DSC measurement. Please specify the equipment which you use to perform XDR measurement. Why you perform your DSC from 40 ºC In order to avoid reader confusion please remove the stabilization period from DSC curves. Please used the common standard programs to work with DSC and XDR. Use the same letter for DSC legend inside the graph. Please draw all 4 axis as in more published papers; applied for every Figure. Additionally, in DSC Figure mark the endo process with arrow. Please see related papers. Heating flow is arbitrary units and the values should be remove from the graph. Please avoid double information. You have melting point and crystalization point at Table and in Figure 2. Please graph Figure 3 from 10 to 70º in this way we will better appreciate the changes for each polyethylene. In present way is difficult to distinguish between each curve. In the case of Figure 6 please make the 4 graphs smaller (two in the same column and to in the same line (see different published papers). In this case readers with be easer compare your results obtained for each polypropylene. Make the same changes for Figure 7 and Figure 8. 

The "low density Polyethylene" and "middle density Polyethylene" in the paper have been modified to "low-density Polyethylene", "middle-density Polyethylene".

Manufacturers of all types of polyethylene have already stated that the supplier of the equipment has also been marked with a yellow line.

In references 22, 23, the form of the DSC curve is the same because the polyethylene melting temperature and crystallization temperature are between 40 and 180 temperatures.

The endo process in the DSC image has been marked.

Heating flow is arbitrary units and the values have been removed from the graph.

Figure 6, Figure 7, and Figure 8 are respectively reduced and re-typed.

The XRD experimental curve in Figure 3, the peak value of the curve of different density polyethylene has been marked in the graph, which is helpful for the reader to visually analyze.

Figure 3. The XRD curves of the four polyethylenes are drawn separately.

Round 2

Reviewer 1 Report

Authors have sincerely responded to the reviewer's suggestions. There are still editorial issues but the manuscript is acceptable. 
